# Packing Technique with or without Remodeling for Endovascular Coil Embolization of Renal Artery Aneurysms: Safety, Efficacy and Mid-Term Outcomes

**DOI:** 10.3390/jcm10020326

**Published:** 2021-01-17

**Authors:** Grégory Secco, Olivier Chevallier, Nicolas Falvo, Kévin Guillen, Pierre-Olivier Comby, Christiane Mousson, Nabil Majbri, Marco Midulla, Romaric Loffroy

**Affiliations:** 1Department of Vascular and Interventional Radiology, Image-Guided Therapy Center, François-Mitterrand University Hospital, 14 Rue Paul Gaffarel, BP 77908, 21079 Dijon, France; gregory.secco@gmail.com (G.S.); olivier.chevallier@chu-dijon.fr (O.C.); nicolas.falvo@chu-dijon.fr (N.F.); kguillen@hotmail.fr (K.G.); marco.midulla@chu-dijon.fr (M.M.); 2Department of Neuroradiology and Emergency Radiology, François-Mitterrand University Hospital, 14 Rue Paul Gaffarel, BP 77908, 21079 Dijon, France; pierre-olivier.comby@chu-dijon.fr; 3Department of Nephrology and Renal Transplantation, François-Mitterrand University Hospital, 14 Rue Paul Gaffarel, BP 77908, 21079 Dijon, France; christiane.mousson@chu-dijon.fr (C.M.); nabil.majbri@chu-dijon.fr (N.M.)

**Keywords:** aneurysm, embolization, endovascular treatment, renal artery

## Abstract

The endovascular treatment of renal artery aneurysms (RAAs) has lower morbidity and shorter stay lengths compared to surgical repair. Here, we describe coil packing with or without remodeling and assess outcomes and complications. We retrospectively identified the 19 consecutive preventive endovascular RAA coil embolizations done in 18 patients at our center in 2010–2020. Patient and aneurysm characteristics, technical success rate, complications, and recurrences were recorded. Mean patient age was 63 ± 13 years. The RAA was >1.5 cm in 11 cases, and in four cases, the aneurysm-to-parent artery size ratio was >2. Simple coiling was performed for 11 (57.9%) aneurysms, stent-assisted coiling for seven (36.8%) aneurysms, and balloon-assisted coiling for one (5.3%) aneurysm. Technical success rate was 100%. Complete definitive RAA exclusion was achieved with a single procedure for 17 (89.5%) aneurysms, whereas two (10.5%) aneurysms required a repeat procedure. Four minor complications occurred but resolved with no long-term consequences. No major complications occurred during the mean follow-up of 41.1 ± 29.7 months. Coil embolization by sac packing or remodeling proved very safe and effective. Together with the known lower morbidity and shorter stay length compared to open surgery, these data indicate that this endovascular procedure should become the preventive treatment of choice for RAAs.

## 1. Introduction

Renal artery aneurysms (RAAs) are rare, with an incidence in the general population of 0.1 to 1%. However, they account for 25% of all visceral artery aneurysms [1]. The most dreaded complication of RAA is rupture with massive bleeding. The risk of rupture increases with the aneurysm diameter and is also higher in specific patient subgroups such as women of childbearing age. Although no strong recommendation exists concerning the indications for RAA treatment, symptomatic aneurysms and aneurysms larger than 1.5 cm are usually deemed to require intervention [1,2,3,4,5,6]. Multiple clinical studies have assessed the efficacy and safety of endovascular treatment versus surgical repair [7,8,9,10]. The choice between these two options is generally based on the location and morphology of the aneurysm. Both techniques are known to be safe and effective. However, endovascular repair is associated with lower morbidity and a shorter length of stay. Various endovascular techniques are used depending on the location and morphology of the aneurysm and on the number of relevant branches arising from the aneurysm. Thus, simple coiling embolization is used for narrow-necked aneurysms. For wide-necked or complex aneurysms, the options include stent- or balloon-assisted coiling and exclusion by stent-grafts. Neuroendovascular techniques have been applied to peripheral aneurysms. Coils that detached instead of having to be pushed, which were developed in 1990 by Guglielmi [11], constituted a breakthrough. The use of detachable microcoils, which are repositionable, allows extraordinary precision in placement, thus allowing effective embolization of aneurysms of different sizes. The balloon remodeling technique was introduced by Moret et al. [12] in 1997 for the treatment of wide-neck intracranial aneurysms: an inflatable balloon is used to avoid coil migration in the parent artery during the procedure and to increase coil density in the aneurysm. Other new tools and techniques allow the endovascular treatment of wide-neck and fusiform aneurysms, while preserving vessel patency. Stent-assisted remodeling is widely used in neuroradiology. It was first described for the treatment of RAAs in 2008 by Manninen et al. [13,14,15]. More recently, flexible stent grafts and flow diverters have been introduced. Since their approval as an innovative stent system for peripheral aneurysm management, multilayer stents have been applied in several clinical cases [16,17]. The unique design of multilayer stents decreases mean velocity and vorticity within the aneurysm sac immediately and causes thrombus to form, resulting in physiological exclusion of the aneurysm from the circulation, whereas branches and collaterals sprouting from the aneurysm remain patent [17]. Few studies are available reporting results of multilayer stents for renal artery aneurysms [18,19]. The purpose of this single-center retrospective study was to specifically describe the procedure and outcomes of coiling with or without remodeling used to treat 19 RAAs in 18 patients over a 10-year period.

## 2. Materials and Methods

### 2.1. Study Population

All procedures for the preventive endovascular treatment of RAAs were identified by searching our university hospital electronic database over a 10-year period (2010–2020). We then selected those patients managed by endovascular coil packing with or without remodeling. Wide necks were defined as a neck diameter greater than 4 mm or as a dome-to-neck ratio less than 2. Patients with post-traumatic pseudoaneurysms or arteriovenous malformations were excluded. Ethical review and approval were waived for this study, in compliance with French legislation on retrospective studies of anonymized data.

### 2.2. Endovascular Procedure

All aneurysms were treated by two interventional radiologists with 15 and 10 years of experience, respectively, under local anesthesia, using the transfemoral approach with the Seldinger technique. The common femoral artery was accessed under ultrasound guidance. After positioning of the femoral sheath, an intravenous bolus of 5000 IU heparin was delivered. Aortography was performed, and the renal artery was accessed with a 5-Fr catheter inserted through a 6-Fr guiding sheath. A 2.4-Fr Progreat microcatheter (Terumo, Tokyo, Japan) was then advanced into the aneurysm sac to allow placement of the coils. Multiple detachable Concerto microcoils (Medtronic, Minneapolis, MN, USA) of decreasing size were placed into the aneurysm sac. Stent-assisted coil embolization was performed using the EverFlex nitinol self-expanding stent (Medtronic, Minneapolis, MN, USA) deployed over a 0.035-inch guidewire. For 1 patient, a Leo nitinol self-expanding braided stent (Balt, Montmorency, France) was used over a 0.014-inch guidewire to accommodate the unusual arterial anatomy. The Leo stent, designed for treating intracranial aneurysms, is resheathable and repositionable and has a high radial force that enables safe and effective use in twisting arteries. In case of stent-assisted remodeling technique, catheterization of the aneurysm sac was performed through the mesh of the stent using a 2.0-Fr microcatheter for coiling of the sac. Balloon-assisted coil embolization was performed by double femoral approach using the Rx Viatrac 14 Peripheral Dilatation Catheter (Abbott, Chicago, IL, USA) of appropriate size. Patients managed with simple coiling or remodeling using a balloon were discharged with no antiplatelet therapy. When a bare stent was used for remodeling, patients were discharged with dual antiplatelet therapy (acetylsalicylic acid, 100 mg daily; clopidogrel, 75 mg daily) for 6 months, after which the clopidogrel was stopped, and the acetylsalicylic acid continued in the same dosage.

### 2.3. Angiographic Outcomes, Complications, and Follow-Up

The final angiographic results were assessed using the modified Raymond–Roy occlusion classification (RROC), which is widely accepted for evaluating the occlusion of intracranial aneurysms [20]. Class 1 is defined as complete obliteration of the sac, Class 2 as subtotal obliteration with a residual neck, and Class 3 as residual aneurysm. Subclass 3a is defined as contrast opacification within the coil interstices of a residual aneurysm and Subclass 3b as contrast opacification outside the coil interstices, along the residual aneurysm wall. All patients underwent follow-up computed tomography (CT) or magnetic resonance imaging (MRI) at least 1 month after the procedure and a final CT or MRI at the end of the study period, in January 2020. The type of cross-sectional imaging was chosen based on the presence of a stent. In that case, MRI was preferred. CT and MRI were performed after intravenous contrast injection with standard protocols. Complete or incomplete occlusion of the sac was noted. Kidney function was evaluated based on the serum creatinine concentration (µmol/L) preoperatively, 1 month after the procedure and, at last, follow-up. Perioperative complications were defined as those occurring during the procedure and up to 1 month after the procedure. They were divided into minor and major complications according the Society of Interventional Radiology classification system [21].

### 2.4. Statistical Analysis

Descriptive statistics and parameters, such as frequencies and percentages, were used and provided in order to accurately describe our experience regarding the endovascular procedure with or without remodeling. Values were presented as means ± SDs (range) for all variables.

## 3. Results

### 3.1. Patients

We identified 18 patients (15 females) with 19 true RAAs (Table 1). Mean age was 63 ± 13 years (range, 39–85). Seventeen patients had a single aneurysm and one patient had two aneurysms on the right renal artery (Figure 1). The circumstances of the diagnosis are shown in Table 2. A history of fibromuscular dysplasia (FMD) was noted in two (11.1%) females and a history of hypertension in six (33.3%) patients (five females and one male). An aneurysm was found on the splenic artery in one (5.6%) patient. The patient with two RAAs had a history of liver transplantation.

### 3.2. Aneurysms

Ten (52.6%) aneurysms were located on the right renal artery and nine on the left renal artery. Mean aneurysm size was 15.7 ± 3.6 mm (range, 10–23 mm) with 11 aneurysms larger than or equal to 15 mm. Mean aneurysm neck size was 6.6 ± 2.7 mm (range, 3–15 mm). Fifteen (79%) lesions were narrow-necked aneurysms. Mean parent artery size was 4.7 ± 1.4 mm (range, 3–8 mm). Seven (36.8%) aneurysms had one efferent branch, and 12 had two efferent branches. Sixteen (84.2%) aneurysms were located on the hilar segment of the renal artery, one (5.3%) at the ostium, one on the trunk, and one on the intra-renal segment. Table 2 lists the indications for treatment: size larger than or equal to 1.5 cm in 11 cases; aneurysm-to-parent artery size ratio >2 in 15 cases; flank pain in one patient; hematuria in two patients; vascular steal syndrome in one patient due to high flow in the aneurysmal sac, and progressiveness in one patient.

### 3.3. Coil Embolization and Outcomes

Simple coiling was performed for 11 (57.9%) aneurysms, stent-assisted coiling for seven (36.8%) aneurysms, and balloon-assisted coiling for one (5.3%) aneurysm. The patient with two aneurysms on the same artery was treated in two sessions and required simple coiling for one aneurysm (Figure 2) and stent-assisted coiling for the other (Figure 3). On the posttreatment angiogram, 18 aneurysms (94.7%) were totally occluded. All aneurysms treated by simple coiling were totally occluded. Of the eight aneurysms treated by coiling with remodeling, one (12.5%) had a residual neck and was graded RROC Class 2. This aneurysm was completely occluded in the short-term and on the final MRI scans. On the short-term MRI scan, one (5.3%) aneurysm was incompletely occluded. This aneurysm was located at the ostium of the right renal artery. Stent-assisted coiling was used initially. Reintervention was with a BeGraft^®^ peripheral covered stent (Bentley, HechinGen, Germany). The final MRI showed complete exclusion of the aneurysm. At the end of the study, mean follow-up was 41.1 ± 29.7 months (range, 1–99). No patients were lost to follow-up. On the final MRI scan, one (5.3%) aneurysm was incompletely occluded. This aneurysm was located at the hilum of the right renal artery and had initially been treated by simple coiling. Reintervention using stent-assisted coiling achieved complete occlusion of the sac. Thus, in all, two (10.5%) aneurysms required reintervention, which achieved complete obliteration.

### 3.4. Complications

As shown in Table 3, the technical success rate was 100%, with no major complications (Figure 4). Two patients experienced perioperative lumbar pain due to renal microinfarcts with no consequence, and one patient experienced mild decompensation of heart failure following the procedure. One patient had transient renal failure consisting in a serum creatinine increase from 83 µmol/L preoperatively to 131 µmol/L immediately after the procedure; the serum creatinine concentration was normal (86 µmol/L) 1 month later. No delayed complications were recorded during follow-up, especially no procedure-related renal impairment (Table 4).

## 4. Discussion

In our study of 19 RAAs in 18 patients managed by endovascular coil embolization, the technical success rate was 100% and no major complications occurred at any point during follow-up. Minor complications in four patients resolved rapidly. Only two aneurysms were incompletely occluded, one shortly after the procedure and the other at the final evaluation. Both aneurysms were successfully treated by endovascular reintervention.

In our population, females predominated (83.3%), in keeping with earlier reports [2,22,23,24,25]. The mean age of about 60 years is also consistent with the literature [1,2,22,23]. Fibromuscular dysplasia is a known cause of RAA and was present in two of our patients. Another known cause of RAA is hypertension with atherosclerosis, which was present in six of our patients [26,27,28,29,30,31,32,33]. However, RAAs can also cause hypertension by embolism to the distal parenchyma or kinking of the renal artery [2]. Patient selection for RAA repair remains controversial, except when bleeding has occurred. Bleeding is the most dreaded complication, and the goal is to repair aneurysms at high risk for bleeding. There is general agreement that repair is in order irrespective of aneurysm size in patients with symptoms (hypertension, flank pain, hematuria), complications (dissection, distal embolization, vascular steal syndrome), or rapid enlargement [34,35,36,37,38,39]. Cases of these indications were represented in our population. In women of childbearing potential, repair is also recommended regardless of aneurysm size. Patients with abnormal blood vessels such as those with Ehlers–Danlos syndrome type IV should also undergo repair given the very high risk of rupture. Larger size is a risk factor for bleeding, and it has been suggested that aneurysms larger than 1.5 cm [1,2,4,5] or 2 cm [7,9,10] should be repaired. We chose the lower cutoff. The ratio of aneurysm size over parent artery size is also important to consider. This criterion has been chiefly studied for intracranial aneurysms and found to be associated with rupture when greater than 2 cm [40,41,42]. We repaired aneurysms with a ratio greater than 2 even when aneurysm size was less than 1.5 cm. Although surgery remains the conventional treatment of RAAs [2,7], endovascular interventions are available for treating all aneurysm types.

Endovascular techniques have both gained in popularity and benefited from considerable technical advances over the last two decades. Compared with open surgery, endovascular repair is associated with a significantly lower rate of postoperative complications and a shorter length of stay [9,10]. A key aspect in the endovascular treatment of RAAs is preservation of kidney vascularization, which is terminal, with no anastomoses between segmental branches. Occlusion of proximal arteries, therefore, results in segmental infarction, which may impair renal function. Therefore, the parent arteries and distal flow must be preserved during endovascular treatment. The endovascular treatment of aneurysms has made huge strides over the last two decades. Neuroendovascular techniques have been applied to peripheral aneurysms, as previously described. The balloon remodeling technique is usually performed for the treatment of wide-neck aneurysms. We used this technique in one patient. Stent-assisted remodeling can also be used. We used this method for seven aneurysms. A bare open-cell stent is positioned, and the microcatheter and microcoils are introduced through the mesh of the stent. The kissing-stent or kissing-balloon remodeling technique is used for aneurysms located at bifurcations, which is often the case with renal arteries: a kissing stent was used for one aneurysm in our population. In one patient, we used a Leo intracranial stent, which is a nitinol self-expanding braided stent that is well suited for aneurysms on small tortuous arteries. Finally, stent grafts and flow diverters can be useful to preserve distal vascularization [16,17,18,19]. Initially, coronary stent grafts were used because of their high flexibility [43]. These balloon-expandable stent grafts are mounted on very thin microcatheters and can reach distal aneurysms. However, coronary stent grafts were available in limited lengths and diameters. Dedicated peripheral stent grafts are now available. An interesting point in our study is the use of simple coiling in most patients. The advantage of this technique is that no antiplatelet therapy is needed after the procedure. Thus, it is particularly recommended in young patients and in women of childbearing age. This technique proved very effective and safe, with no major complication and only one recurrence [44,45]. If needed, reintervention by endovascular techniques is usually simple. The effectiveness and safety of endovascular embolization to treat RAAs has been the focus of many studies since 1995 [7,8,9,30,32]. In early publications, effectiveness was good, but the complication rate remained high, due chiefly to the intrinsic limitations of the material available at the time [46,47]. Studies done in the last decade to assess the endovascular treatment of 18 and 52 aneurysms, respectively [32,48], used a variety of endovascular techniques including coiling, balloon and stent-assisted coiling, coil trapping, and parent artery occlusion and recorded no major complications. Our patients also had no major complications. There were four transient minor complications. In addition, renal microinfarcts with no consequence on renal function were seen in two patients. No delayed complications were observed during follow-up. Two recurrences were successfully treated using endovascular techniques. Some other embolic agents can be used alone or in combination with coils for the treatment of RAAs. Indeed, liquid embolic agents such as cyanoacrylates or ethylene vinyl alcohol copolymers may be of interest in order to completely fill in the aneurysmal sac after partial coiling, avoiding the use of too many coils. This combined technique can be very useful around the end of the sac packing with coils thanks to the ease of liquid injection through the microcatheter, whereas packing is not possible anymore. Liquid injection into the sac with balloon remodeling technique may even be used in selected aneurysms in well-trained hands. [49,50,51].

Our study has the limitations inherent in its retrospective design. In addition, over the 10-year recruitment period, technical advances may have occurred, and experience was accumulated by the operators. The main limitation is probably the small sample size, with only 19 aneurysms in 18 patients. However, with such a sample, our study represents one of the largest series reported in the literature, due to the rarity of this condition. Follow-up was only about 3 years. Nevertheless, the consistency of the data with a very high success rate and both recurrences successfully treated by endovascular techniques support this treatment method for patients with RAAs [52]. Few experimental in vitro and/or in vivo studies are reported regarding the use of numerical simulation and laboratory modelling to assess the different endovascular techniques available for the treatment of renal artery aneurysms [16,17,53]. Evaluation of functionality and biological response of those models would be of major interest and needs further studies.

## 5. Conclusions

Technical advances have improved the efficacy and safety of endovascular techniques for the treatment of RAAs, even in comparison with standard surgical repair. In addition, endovascular repair is less invasive and aggressive than surgery. Endovascular treatment deserves to be widely used for the preventive repair of asymptomatic RAAs.

## Figures and Tables

**Figure 1 jcm-10-00326-f001:**
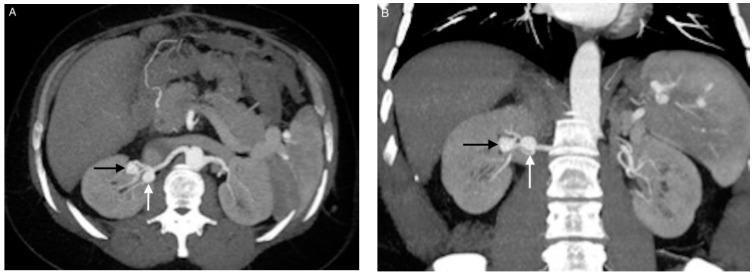
Images from patient 6. (**A**) and (**B**) Computed tomography angiography showing aneurysms 6 (white arrow) and 7 (black arrow) at the hilum of the right kidney. Aneurysm 6 is located at the initial trifurcation of the renal artery and has a wide neck. Aneurysm 7 is more distal, at a secondary bifurcation of the renal artery, and has a narrow neck.

**Figure 2 jcm-10-00326-f002:**
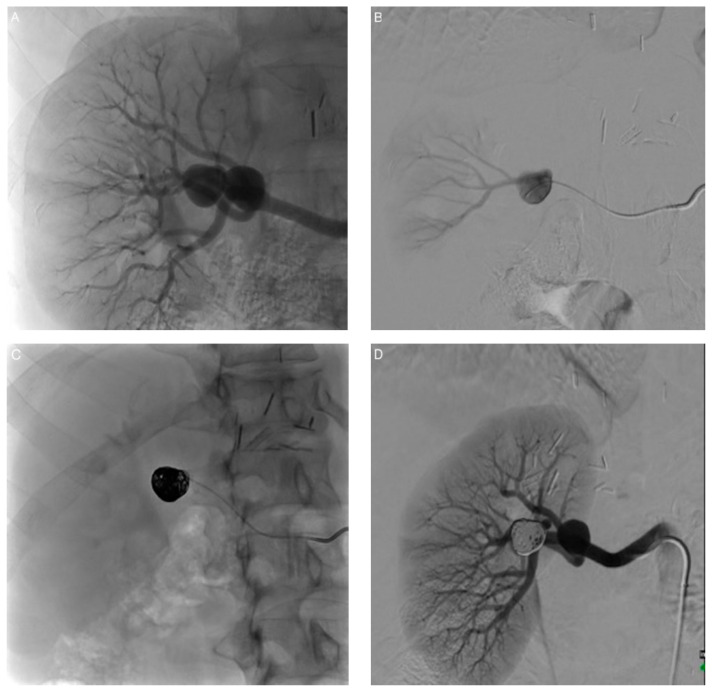
Exclusion of aneurysm 7 by packing embolization. (**A**) Angiogram revealing the two aneurysms. (**B**) Preliminary angiogram performed after selective aneurysm sac catheterization. (**C**) Coil embolization. The aneurysm is fully packed with detachable microcoils. (**D**) After coil embolization, renal artery angiogram shows complete obliteration of the sac with no residual flow or neck.

**Figure 3 jcm-10-00326-f003:**
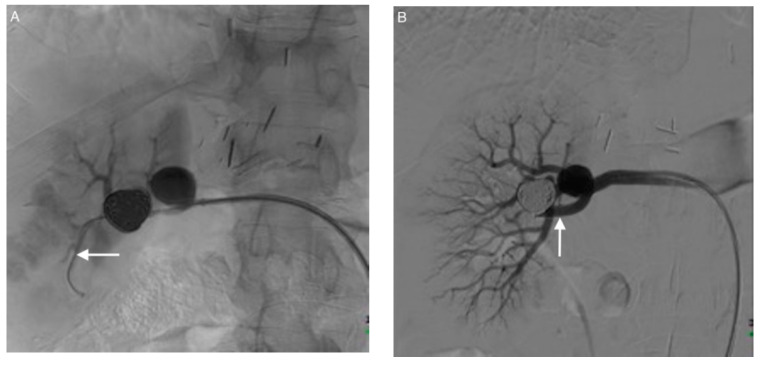
Exclusion of aneurysm 6 in a second session: stent-assisted coiling. (**A**) A guidewire is navigated beyond the aneurysm into an inferior segmental branch (arrow). (**B**) A bare stent (arrow) is deployed over the aneurysm neck, extending from the main renal artery to the inferior branch of the renal artery. (**C**) Catheterization of the aneurysm sac through the mesh of the stent using a microcatheter (arrow) and coiling of the sac. (**D**) Angiography performed at the end of the procedure showing complete exclusion of both aneurysms.

**Figure 4 jcm-10-00326-f004:**
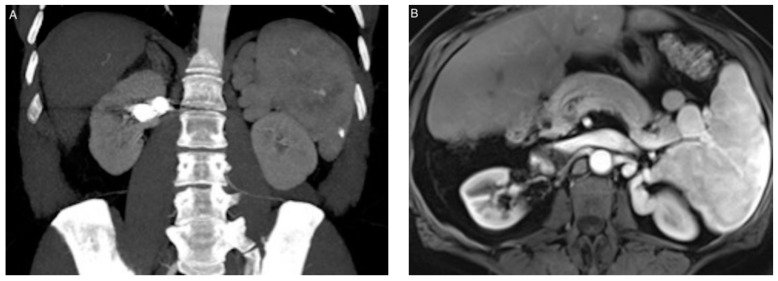
Follow-up computed tomography angiography at 1 month (**A**) and magnetic resonance imaging at 36 months (**B**) showing complete exclusion of both aneurysms, whose sizes remain unchanged.

**Table 1 jcm-10-00326-t001:** Characteristics of the 18 patients and 19 aneurysms.

Patient/ Aneurysm	Age (years)	Sex	Side (Right–Left)	Size (mm)	Neck Size (mm)	Parent Artery Size (mm)	Number of Efferences	Location
1/1	83	Female	R	23	15	6	1	Ostial
2/2	55	Female	L	11	6	5	1	Hilar
3/3	58	Female	R	15	6	4	2	Hilar
4/4	60	Female	R	13	4	3	2	Hilar
5/5	85	Female	L	18	5	3	2	Intrarenal
6/6	39	Female	R	15	9	5	1	Hilar
6/7	39	Female	R	13	5	5	1	Hilar
7/8	54	Female	R	20	7	6	2	Hilar
8/9	44	Female	L	21	6	6	2	Hilar
9/10	79	Female	R	13	3	3	1	Hilar
10/11	62	Female	L	14	9	3	2	Hilar
11/12	55	Female	R	14	6	4	2	Hilar
12/13	65	Female	L	16	7	5	2	Hilar
13/14	57	Female	L	17	6	4	1	Truncal
14/15	66	Male	L	10	5	5	1	Hilar
15/16	70	Female	L	12	5	5	2	Hilar
16/17	71	Male	R	18	9	6	2	Hilar
17/18	78	Male	L	20	8	8	2	Hilar

**Table 2 jcm-10-00326-t002:** History, clinical presentation, and indications for treatment.

Patient/Aneurysm	History	Clinical Presentation	Indications for Treatment
1/1	Hypertension	Incidental discovery	Size ≥ 15 mm
2/2	Hypertension	Vascular steal syndrome	Symptomatic aneurysm
3/3	None	Incidental discovery	Size ≥ 15 mm
4/4	None	Hematuria	Symptomatic aneurysm
5/5	Hypertension	Assessment of hypertension	Renovascular hypertension, Size ≥ 15 mm
6/6	Liver transplantation	Incidental discovery	Size ≥ 15 mm
6/7	Liver transplantation	Incidental discovery	Aneurysm-to-parent artery size ratio > 2, Treatment of another RAA
7/8	Hypertension	Flank pain	Symptomatic aneurysm
8/9	Fibromuscular dysplasia	Incidental discovery	Size ≥ 15 mm
9/10	None	Incidental discovery	Aneurysm-to-parent artery size ratio > 2
10/11	Hypertension	Assessment of hypertension	Renovascular hypertension
11/12	None	Incidental discovery	Size ≥ 15 mm
12/13	None	Incidental discovery	Size ≥ 15 mm
13/14	None	Incidental discovery	Size ≥ 15 mm
14/15	None	Incidental discovery	Aneurysm-to-parent artery size ratio > 2
15/16	None	Incidental discovery	Aneurysm-to-parent artery size ratio > 2
16/17	None	Incidental discovery	Progressiveness
17/18	Hypertension	Hematuria	Symptomatic aneurysm

**Table 3 jcm-10-00326-t003:** Technique, angiographic results, and complications.

Patient/ Aneurysm	Endovascular Technique	Postoperative Angiographic Results (RROC)	Perioperative Minor Complications	Perioperative Major Complications
1/1	Stenting + coiling	Class 1	None	None
2/2	Coiling	Class 1	Pain	None
3/3	Coiling	Class 1	None	None
4/4	Balloon + coiling	Class 1	Pain	None
5/5	Coiling	Class 1	None	None
6/6	Stenting + coiling	Class 1	None	None
6/7	Coiling	Class 1	None	None
7/8	Coiling	Class 1	None	None
8/9	Coiling	Class 1	None	None
9/10	Coiling	Class 1	None	None
10/11	Stenting + coiling	Class 1	Mild decompensated heart failure	None
11/12	Coiling	Class 1	None	None
12/13	Coiling	Class 1	None	None
13/14	Coiling	Class 1	None	None
14/15	Stenting + coiling	Class 1	None	None
15/16	Stenting + coiling	Class 1	None	None
16/17	Stenting + coiling	Class 2	Transient renal failure	None
17/18	Coiling	Class 1	None	None

RROC, Raymond–Roy occlusion classification.

**Table 4 jcm-10-00326-t004:** Short- and long-term follow-up results.

Patient/ Aneurysm	Short-Term Imaging Results	Long-Term Imaging Results	Duration of Follow-Up (months)
1/1	Incomplete	Complete occlusion after reintervention	99
2/2	Complete	Complete	79
3/3	Complete	Complete	78
4/4	Complete	Complete	59
5/5	Complete	Complete	53
6/6	Complete	Complete	40
6/7	Complete	Complete	40
7/8	Complete	Incomplete–complete occlusion after reintervention	25
8/9	Complete	Complete	24
9/10	Complete	Complete	21
10/11	Complete	Complete	21
11/12	Complete	Complete	20
12/13	Complete	Complete	15
13/14	Complete	Complete	93
14/15	Complete	Complete	31
15/16	Complete	Complete	36
16/17	Complete	Complete	42
17/18	Complete	-	1

## Data Availability

The data presented in this study are available on request from the corresponding author. The data are not publicly available due to identity reasons.

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
