# Peer review of "Packing Technique with or without Remodeling for Endovascular Coil Embolization of Renal Artery Aneurysms: Safety, Efficacy and Mid-Term Outcomes"

_jcm, 2021, doi:10.3390/jcm10020326_

Round 1

Reviewer 1 Report

The paper by Secco et al. described the endovascular treatment of renal artery aneurysms; it reads well and results are clearly presented and discussed.

Before accepting it for the publication on JCM, I would suggest to improve it following the indication below.

INTRODUCTION

The introduction need to be more detailed, I noticed that in your References don’t include more recent research work as Scientific Reports | (2019) 9:10193 | https://doi.org/10.1038/s41598-019-46714-7.

In addition, if in this context you are aware of in vitro studies and/or studies based on numerical simulations, they have to be included and discussed in relation to your work/findings.

SECTION 2

Line 91: is there an analogue of ROOC as for RAs?

DISCUSSION

Line 196: check the parenthesis after ref 24;

Lines 222 (from Neuroendovascular) to 232: should be moved in the Introduction, being (a part of) the State of the Art;

Lines 258-265: as for the limitation of the presented study (rightly highlighted), don’t the Authors think that performing numerical and /or laboratory modelling of this problem may help in this sense? Eventually add a comment and the appropriate references.

Author Response

Reviewer 1

  1. English language and style are fine/minor spell check required. 

Reply: Thanks for this comment. English language editing has been performed by a native speaker.

  1. The paper by Secco et al. described the endovascular treatment of renal artery aneurysms; it reads well and results are clearly presented and discussed.

Reply: Thanks for this comment.

  1. Before accepting it for the publication on JCM, I would suggest to improve it following the indication below.

Reply: Thanks for this comment. The paper has been improved as suggested.

  1. Introduction: The introduction needs to be more detailed, I noticed that in your References don’t include more recent research work as Scientific Reports | (2019) 9:10193 | https://doi.org/10.1038/s41598-019-46714-7. In addition, if in this context you are aware of in vitro studies and/or studies based on numerical simulations, they have to be included and discussed in relation to your work/findings.

Reply: Thanks for this comment. The introduction has been more detailed as suggested. Some specific references below have been included in this section and discussed in the limitation paragraph of the discussion accordingly as suggested.

Li, Z.; Hu, L.; Chen, C.; Wang, Z.; Zhou, Z.; Chen, Y.Hemodynamic performance of multilayer stents in the treatment of aneurysms with a branch attached.Sci Rep2019, 9, 10193, doi: 10.1038/s41598-019-46714-7.

Alherz, A.I.; Tanweer, O.; Flamini, V.J. A numerical framework for the mechanical analysis of dual-layer stents in intracranial aneurysm treatment.Biomech2016, 49, 2420–2427, doi: 10.1016/j.jbiomech.2016.02.026.

Balderi, A.; Antonietti, A.; Pedrazzini, F.; Sortino, D.; Vinay, C.; Grosso M. Treatment of visceral aneurysm using multilayer stent: two-year follow-up results in five consecutive patients. Cardiovasc Intervent Radiol2013, 36, 1256–1261, doi: 10.1007/s00270-013-0705-0.

Meyer, C.; Verrel, F.; Weyer, G.; Wilhelm, K. Endovascular management of complex renal artery aneurysms using the multilayer stent. Cardiovasc Intervent Radiol2011, 34, 637–641, doi: 10.1007/s00270-010-0047-0.

  1. Section 2: Line 91: is there an analogue of ROOC as for RAs?

Reply: Thanks for this comment. Unfortunately there is not. The reference for occlusion of aneurysms comes from the RROC classification which is the gold standard.

  1. Discussion: Line 196: check the parenthesis after ref 24.

Reply:Thanks for this comment. It has been corrected.

  1. Discussion: Lines 222 (from Neuroendovascular) to 232: should be moved in the Introduction, being (a part of) the State of the Art.

Reply:Thanks for this comment. This part has been moved to the introduction section as suggested. All references have been renumbered consequently.

  1. Discussion: Lines 258-265: as for the limitation of the presented study (rightly highlighted), don’t the Authors think that performing numerical and /or laboratory modelling of this problem may help in this sense? Eventually add a comment and the appropriate references.

Reply:Thanks for this comment. For sure, numerical and/or laboratory modelling would be of major interest for better understanding. It has bene added in the limitations section with appropriate references as suggested, especially the following reference. (Sultan, S.; Kavanagh, E.P.; Hynes, N.; Diethrich, E.B. Evaluation of functionality and biological response of the multilayer flow modulator in porcine animal models.Int Angiol2016, 35, 31–39). This reference has no DOI available.

Reviewer 2 Report

the presented group is quite large and well described, it is a pity that it is not a prospective study with a control group.

Author Response

  1. English language and style are fine/minor spell check required. 

Reply: Thanks for this comment. English language editing has been performed by a native speaker.

  1. The presented group is quite large and well described, it is a pity that it is not a prospective study with a control group.

Reply: Thanks for this comment. We fully agree with this comment. Unfortunately, this is a retrospective study. There is no control group in our study. We reported our experience with this technique retrospectively given the rarity of this condition.

Reviewer 3 Report

Good work, only some minor issues:

 - Page 3 Lines 109-110: what about non normal distributed data?

 - P 4 LL 129-130: "Wide 129 necks [...] than 2." --> Move to M&M

 - Table 1 and 2. Patient 2 aneurysm has only one efference. How can only one efference aneurysm entail vascular steal syndrome? Please explain for non expert readers.

Author Response

1. Good work, only some minor issues.

Reply: Thanks for this comment. Minor issues have been figured out.

2. Page 3 Lines 109-110: what about non-normal distributed data?

Reply: Thanks for this comment. There were no non-normal distributed data in our paper. It has been modified in the corresponding section.

3. P 4 LL 129-130: "Wide 129 necks [...] than 2." --> Move to M&M

Reply: Thanks for this comment. As suggested, this sentence has been moved to the study population paragraph of the materials and methods section.

4. Tables 1 and 2. Patient 2 aneurysm has only one efference. How can only one efference aneurysm entail vascular steal syndrome? Please explain for non expert readers.

Reply:Thanks for this comment. The clinical presentation in this patient was arterial hypertension. The cause of this hypertension was attributed to the aneurysm by vascular steal syndrome. It is always difficult to make clear the causality but this is what was suspected here clinically. The supposed mechanism is the high flow inside the aneurysmal sac leading to steal syndrome on the angiography. It has been added in the text in the aneurysms paragraph of the results section.